# A rapid and easy-to-use spinal muscular atrophy screening tool based on primers with high specificity and amplification efficiency for *SMN1* combined with single-stranded tag hybridization assay

Masaki Hirano[1,2], Kentaro Sahashi[3,4], Yuki Ichikawa[2], Masahisa Katsuno[3,4,5]*, Atsushi Natsume[5,6]*

**1** Division of Molecular Oncology, Aichi Cancer Center Research Institute, Nagoya, Aichi, Japan, **2** Craif Inc., Tokyo, Japan, **3** Department of Clinical Research Education, Nagoya University Graduate School of Medicine, Nagoya, Aichi, Japan, **4** Department of Neurology, Nagoya University Graduate School of Medicine, Nagoya, Aichi, Japan, **5** Institute of Innovation for Future Society of Nagoya University, Nagoya, Aichi, Japan, **6** Kawamura Medical Associates, Nagoya, Aichi, Japan

* atsushi.natsume@mirai.nagoya-u.ac.jp (AN); ka2no@med.nagoya-u.ac.jp (MK)

## Abstract

Spinal muscular atrophy (SMA) is an intractable neuromuscular disorder primarily caused by homozygous deletions in exon 7 of the *SMN1* gene. Early diagnosis and prompt treatment of patients with SMA have a significant impact on prognosis, and several therapies have recently been developed. Current SMA screening tests require a significant turnaround time to identify patients with suspected SMA, due both to the interval between the birth of a newborn and the collection of blood for newborn mass screening and the difficulty in distinguishing between *SMN1* and *SMN2*, a paralog gene that requires testing in specialized laboratories. The aim of this study was therefore to develop a novel SMA screening assay that can be rapidly performed in ordinary hospitals and clinics to overcome these issues. We designed over 100 combinations of forward and reverse primers with 3′ ends targeting *SMN1*-specific sites around exon 7, and evaluated their specificity and amplification efficiency by quantitative PCR to identify the best primer pair. Furthermore, we performed a single-stranded tag hybridization assay after PCR. To evaluate the accuracy and practicality of the newly developed assay, we analyzed saliva specimens from five patients with SMA and two SMA carriers collected in an outpatient clinic and DNA specimens from three patients with SMA and four SMA carriers from a biobank, together with those from healthy individuals. DNA and raw saliva specimens from all patients with SMA demonstrated a biallelic loss of *SMN1*, whereas those from carriers and healthy individuals did not. The results of 50 independent experiments were consistent for all samples. The assay could be completed within one hour. This simple and convenient new screening tool has the potential to allow patients with SMA to receive disease-modifying therapies within a shorter timeframe.

**Data Availability Statement:** All relevant data are within the manuscript and its Supporting Information files.

**Funding:** In this study, novel primers were designed and basic performance evaluation of the primers and the STH assay using them was performed in the laboratory of the funder, Craif Inc. To avoid arbitrariness in the clinical trial, anonymous random human saliva samples were processed and tested in the laboratory of Craif and the test results were determined and analyzed by collaborators with no conflict of interest. The funder had no role in clinical trial design, data collection and analysis in the clinical trial, decision to publish, or preparation of the manuscript.

**Competing interests:** I have read the journal's policy and the authors of this manuscript have the following competing interests: Y.I. is a Board member of Craif and owns stock of Craif. M.H. is an employee of Craif and owns stock options of Craif. A.N. has a conflict of interest because he received an honorarium for academic guidance from Craif. Craif is currently applying for a patent for the deliverables developed in this study.

## Introduction

Spinal muscular atrophy (SMA) is an autosomal recessive neuromuscular disorder primarily caused by homozygous deletions in exon 7 of the *survival motor neuron 1* (*SMN1*) gene, leading to lower motor neuron degeneration [1]. In Japan, the incidence has been reported to be 0.51 in 10,000 live births [2], whereas the incidence of pan-ethnic disease is 0.91 in 10,000 [3].

SMA was difficult to treat in the past; however, novel treatments, including antisense oligonucleotides and gene therapy, have recently been developed [4–7]. These therapies increase the production of functional SMN proteins, which rescues motor neurons and alleviates muscle atrophy. However, while nearly normal motor development can sometimes be achieved when treatment is started in infants at a pre-symptomatic stage, therapy is less effective in cases where symptoms have already appeared, making it urgent to establish a method to rapidly detect patients with SMA [8].

Newborn screening tests for SMA are becoming more widespread in Japan, but require highly specialized quantitative PCR (qPCR) instruments and special reagent kits [9, 10]. Consequently, specimens collected from newborns must be stored in specialized laboratories. In addition, dried blood spots are used for these tests; however, it is necessary to allow four to six days between birth and blood collection [9, 10]. These circumstances lead to a significant loss of time until a suspected SMA is detected. Delayed diagnosis can have a serious negative impact on treatment outcomes [8, 11]. Thus, establishing a rapid and easy-to-use test that can be performed in ordinary hospitals or clinics would be beneficial for patients with SMA.

Screening for SMA is difficult due to the similarity of an additional gene, *SMN2*, in addition to the causative gene, *SMN1*, which differs by only 11 bases [1, 12]. Most SMA cases are caused by a homozygous deletion of *SMN1*, including the exon 7 region, which is difficult to detect by conventional PCR assays using standard primers because of the amplification of nonspecific *SMN2*. In this study, we developed a new SMA screening assay that can be performed in ordinary hospitals and clinics which uses new primers with extremely high specificity and amplification efficiency for *SMN1*. Furthermore, we adopted a single-stranded tag hybridization (STH) assay after the PCR and validated its accuracy using genomic DNA and saliva specimens derived from patients with SMA. To the best of our knowledge, this is the first screening assay which detects the homozygous deletion of *SMN1* exon 7 that can be performed in a clinical setting within approximately one hour.

## Methods

### Ethics statement

This study was approved by the Ethics Review Committee of Nagoya University Hospital (approval number:2023–0038), and was conducted in compliance with all of the provisions of the Declaration of Helsinki and the Ethical Guidelines for Medical and Health Research Involving Human Subjects endorsed by the Japanese government.

### Prospective clinical study design

From August 3, 2023, to September 21, 2023, five patients with SMA attending the Department of Neurology, Nagoya University Hospital, and two of their parents were enrolled in this study. All saliva specimens were collected at Nagoya University Hospital after obtaining written informed consent from the patients. The collected saliva was mixed with a preservative solution (Biocomma, Japan), or soaked in medical-grade sponge sheets (Cenefom, Taiwan), anonymized, and transported to the Craif, Inc. laboratory. After performing the *SMN1* copy number (CN) evaluation tests using saliva samples (raw saliva in sponge sheets for the novel

assay and extracted DNA from saliva for the multiple ligation-dependent probe amplification (MLPA) assay), the test results were linked to patient information by collaborators at the Department of Neurology, Nagoya University Hospital, to evaluate the consistency between the results of the novel assay developed in this study and those of the MLPA method, the current gold standard for SMA diagnosis.

## Sample sources

Genomic DNA (gDNA) specimens derived from patients with SMA and carriers were purchased from Coriell Institute (NJ, USA). Saliva specimens were collected from SMA patients attending the Department of Neurology, Nagoya University Hospital, and their parents (SMA carriers), placed on sponge sheets or mixed with a saliva preservation solution, and then transported to the Craif Inc. laboratory, as described above.

As positive control samples, commercially available genomic DNA from a single healthy human adult was purchased from BioChain Institute Inc. (CA, USA). Saliva samples were collected from healthy volunteers.

## DNA isolation from saliva specimens from SMA patients

Genomic DNA was isolated from saliva specimens collected from healthy adults, patients with SMA, and SMA carriers using the QIAmp DNA Mini Kit (Qiagen, Hilden, Germany), according to the manufacturer's instructions.

## Specific primer sets for SMN1 and real-time qPCR

For *SMN1*-specific amplification, we first designed new primer sets, such that one of the five different sites between the *SMN1* and *SMN2* exon 7 peri-regions was included within the third base from the 3′ end of the forward and reverse primers, yielding over 100 combinations in total. Next, we evaluated *SMN1* specificity and amplification efficiency compared with that of the control gene (*RPPH1*) for each primer combination by real-time qPCR. The combination with the best *SMN1* specificity and amplification efficiency in our study was as follows: forward primer: 5′-TTCCTTTATTTTCCTTACAGGGTTCCAG-3′ and reverse primer: 5′-TTGTTT TACATTAACCTTTCAACTTTT-3 (S1 Fig). ' The primer set for *RPPH1* was as follows; forward primer: 5′-CTTTGCCGGAGCTTGGA-3′ and reverse primer: 5′-GAGAGTAGTCTGAATTGGGT TATGA-3. ' We also used other previously reported primers targeting *SMN1* used for SMA screening tests in Germany [13]; forward primer: 5′- TATTTTCCTTACAGGGTTCCAG-3′ and reverse primer: 5′- GCTGGCAGACTTACTCCTTAATTTAA-3 (S1 Fig). '

Real-time qPCR was performed using a QuantStudio 3 system (Life Technologies, CA, USA) with TaqMan Fast Advanced Master Mix (Life Technologies) at least in triplicate, with initial denaturation at 50˚C for 2 min and 95˚C for 10 min, followed by 40 cycles in two steps at 95˚C for 15 s, 60˚C for 1 min. A single acquisition of fluorescence signals was included in the 60˚C step.

The FAM-labeled TaqMan MGB probe (Thermo Fisher Scientific, MA, USA) was used to detect both *SMN1* and *SMN2* in all qPCR experiments of this study; 5′-FAM−ACAAAATCAAA AAGAAGGAAGGTGCTCACA−NFQ−MGB-3′, which had the same sequence (but different reporter dye) as the probe used in the previous report [13]. We also used the NED-labeled TaqMan MGB probe (Thermo Fisher Scientific) to detect *RPPH1*; 5′-NED−ACCTCACCTCAGCC ATTGAACTCAC−NFQ−MGB-3. ' Each reaction mixture (20 μL) contained 1 ng gDNA, 200 nM forward and reverse primers, and 200 nM TaqMan MGB probe. The relative *SMN1* amplification efficiency compared to the control gene (*RPPH1*) for each primer set was calculated using the following formula:

$$\text{Relative amplification efficiency (SMN1)} = 2^{\{\text{Ct(RPPH1)} - \text{Ct(SMN1)}\}}$$

## Conventional non-quantitative PCR

We applied a conventional PCR reaction involving the following steps: 30 cycles with denaturation at 98°C for 10 s, annealing at 62°C for 30 s, and extension at 72°C for 60 s. Each reaction mixture (20 μL) comprised 3 ng of gDNA, forward and reverse primers diluted to 1 μM, and 0.1 μL Taq polymerase (TaKaRa, Japan). The newly designed primers for *SMN1* described above, as well as the previously reported primers used for SMA screening tests in Germany [13] were used to amplify the target region of *SMN1* (target sizes: 200 and 82 bp, respectively). We also used the above-described primers to amplify the target region of *RPPH1*.

## Agarose gel electrophoresis

Each PCR product (10 μL) was dyed with 1 μL of Midori Green (Nippon Genetics, Japan), separated on a 4% agarose gel at 7 V/cm for 60 min with 20 bp DNA ladder (TaKaRa, Japan) or a 2% agarose gel at 7 V/cm for 25 min with Gene Ladder 100 (Nippon Gene, Japan), and then visualized using a FAS-BG LED BOX (Nippon Genetics).

## Single-stranded tag hybridization (STH) assay

We used a single-stranded tag hybridization (STH) assay provided by TBA Inc. (Sendai, Japan) to detect the PCR amplification products without electrophoresis [14]. In this assay, PCR amplification of the target genomic region was performed using the above reverse primers labeled with biotin at the 5′ end and the above forward primers labeled with single-tag DNA at the 5′ end (F-1 tag for SMN1 primer, F-4 tag for RPPH1 primer), which were modified by TBA. The PCR amplification product was then poured onto a membrane strip with avidin (biotin-binding protein)-coated latex (blue) and trapped by a strong hybridization reaction between the complementary tagged DNA immobilized in a linear shape on the strip. The trapped latex-labeled PCR amplification products turned blue, enabling visual identification of the presence or absence of the target genomic region in the specimen. All experiments were repeated at least thrice.

## Multiple ligation-dependent probe amplification (MLPA) assay

All genomic DNA specimens extracted from the saliva of patients with SMA and carriers were tested using the commercially available SALSA MLPA P021 kit for SMA (MRC Holland, Netherlands), according to the manufacturer's instructions. The data were analyzed using Coffalyser software provided by the manufacturer.

## Statistical analysis

The statistical significance of the differences between two groups was analyzed using a paired Student's t-test. All reported P values were two-sided, with $p < 0.05$ considered statistically significant.

# Results

## SMN1 specificity and amplification efficiency of the newly developed primers

First, we assessed the *SMN1* specificity and amplification efficiency of the newly developed primer sets using qPCR. gDNA samples (1 ng) from healthy adults (*SMN1* CN 2), patients

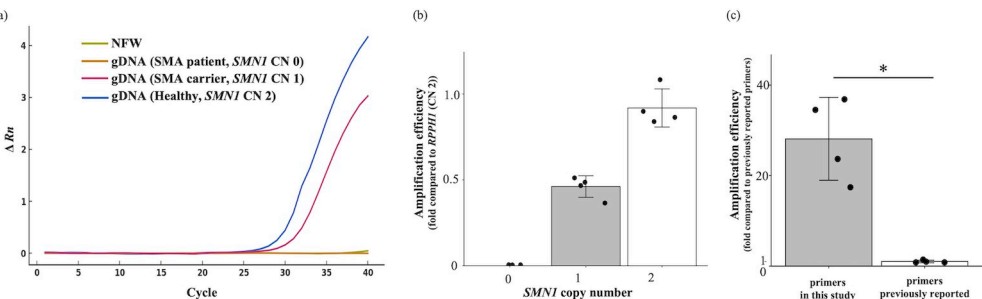

**Fig 1.** *SMN1* **specificity and amplification efficiency of the newly designed primer pairs.** (a) Exemplar normalized FAM fluorescence versus cycle number resulting from *SMN1* amplification of the gDNA of healthy adults, patients with spinal muscular atrophy (SMA), and carriers, using the newly designed primers. A negative control sample (nuclease-free water, NFW) was tested simultaneously. (b) The relative amplification efficiency of *SMN1* (copy number, CN 0, 1, and 2) using the newly designed primers (n = 4, respectively). The fold changes relative to *RPPH1* (CN 2) are shown. (c) Relative amplification efficiency of *SMN1* (CN 2) using the newly designed primers and the previously reported SMA screening primers (n = 4). The fold changes relative to the previously reported primers are shown. Error bars represent ± standard deviation. *p <0.05 (t-test).

with SMA (*SMN1* CN 0), and SMA carriers (*SMN1* CN 1) were amplified in the target region of the *SMN1* and *RPPH1* genes using the different primer pairs, and the combination considered to be the best was used in the following study (S1 File). The gDNA from healthy adults and SMA carriers showed *SMN1* amplification, whereas the gDNA from patients with SMA showed no amplification, which was similar to the negative control sample (nuclease-free water, NFW) (Fig 1A). The relative amplification efficiency of *SMN1* using gDNA (*SMN1* CN 0, 1 and 2) compared to the control gene *RPPH1* (CN 2) was as follows: *SMN1* CN 0:0 ± 0 fold, CN 1:0.46 ± 0.06 fold, CN 2:0.92 ± 0.11 fold (mean ± SD). There was no significant difference in amplification efficiency between *RPPH1* (CN 2) and *SMN1* (CN 2) (Fig 1B).

Furthermore, we compared the amplification efficiency of the *SMN1* target region between our primers (200 bp) and the previously reported primers (82 bp) used for SMA screening in Germany [13, 15, 16]. Although neither primer showed nonspecific *SMN1* amplification, our new primers showed a significantly higher *SMN1* amplification efficiency than the previously reported primers (28.1 ± 9.2 fold, p = 0.0049) (Fig 1C).

## Visibility of PCR products by agarose gel electrophoresis

To develop a simple screening assay, we evaluated the visibility of PCR products by agarose gel electrophoresis. In the *SMN1* amplification assay, neither the primers used in this study, nor the previously reported primers showed any amplified bands with gDNA (*SMN1* CN 0). However, compared with the band of the control gene *RPPH1*, which was simultaneously amplified with the same amount of gDNA in a separate well, the *SMN1* amplification efficiency of the previously reported primers was markedly lower, as suggested by the qPCR results described above (Fig 2A). In the simultaneous *SMN1* and *RPPH1* amplification assays performed using the newly developed primers, no *SMN1* amplification bands were observed for gDNA (*SMN1* CN 0). With other gDNA (*SMN1* CN 1 and 2), both *SMN1* and *RPPH1* were sufficiently amplified. In contrast, in the same assay using previously reported primers, the amplified *SMN1* bands were not detectable, unlike those for *RPPH1* (Fig 2B).

Assuming the contamination of neonatal-derived specimens with maternal tissue, an additional experiment was performed using a mixture of gDNA (*SMN1* CN 0) and gDNA (*SMN1* CN 2) at various ratios. When there was less than 1% contamination of the normal gDNA (*SMN1* CN 2) with gDNA (*SMN1* CN 0), one band of the *SMN1* became almost undetectable (Fig 2C).

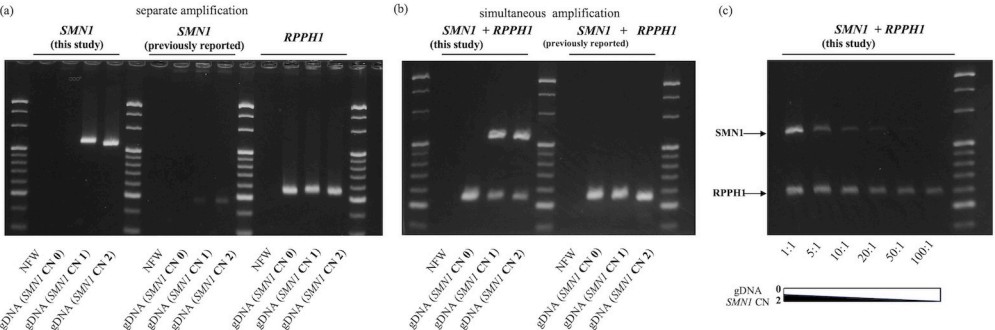

**Fig 2. Visibility of PCR products by agarose gel electrophoresis.** Agarose gel electrophoresis of PCR products using gDNA with *SMN1* copy numbers (CN) of 0, 1, and 2 (with 20 bp ladder). (a) Target regions of *SMN1* (200 or 82 bp) and *RPPH1* (102 bp) were amplified separately using the same amount of input gDNA. (b) Target regions of *SMN1* (200 or 82 bp) and *RPPH1* (102 bp) were simultaneously amplified using the newly designed primers (left) and the previously reported primers (right). (c) Mixtures of gDNA (*SMN1* CN 0) and gDNA (*SMN1* CN 2) at various ratios were amplified under the same conditions as those described in (b). From left to right, the mixing ratios of gDNA (*SMN1* CN 0) and gDNA (*SMN1* CN2) were 1:1, 5:1, 10:1, 20:1, 50:1, and 100:1.

## Establishment of an innovative simple screening assay by combining SMN1-specific primers with an STH assay

To convert the non-quantitative conventional PCR assay described above into a system that can be performed in ordinary hospitals or clinics, we employed the STH assay (TBA, Japan). We confirmed that the same pattern of blue bands, indicating the presence of the target region, was detected in the STH assay using the newly developed primers as in agarose gel electrophoresis. However, the same assay using previously reported primers failed to identify blue bands corresponding to *SMN1* (Fig 3A). To evaluate the effect of gDNA contamination in patients with SMA, STH assays were performed on the same samples as in Fig 2C, and at least 10% contamination of normal gDNA with gDNA from SMA patients did not affect the test results (S2 Fig).

To further simplify the test system, the possibility of performing direct PCR amplification using body fluids (saliva) was investigated. To add a stable amount of saliva to the reaction system, we first soaked a medical sponge sheet (0.8 mm thick) with saliva, punched out a section with a puncher (1.5 mm diameter), and added it to the PCR solution. PCR was then performed, followed by further evaluation using the STH assay. Consequently, we confirmed that

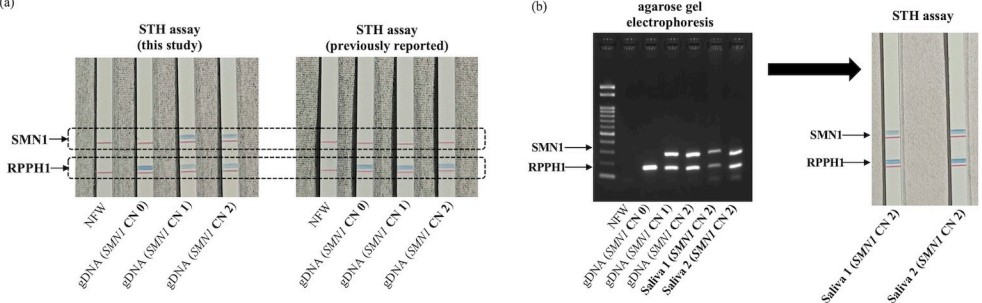

**Fig 3. Establishment of an innovative simple screening assay by combining *SMN1*-specific primers with STH assay.** (a) STH assay results of the same sample set shown in Fig 2B (with primers modified for STH assay). (b) Evaluation of direct PCR products using healthy saliva samples (left: agarose gel electrophoresis with 100 bp ladder; right: STH assay). The blue bands in the STH assay indicate the presence of the target gene region.

the direct PCR method using saliva yielded the same results as the PCR method using the isolated gDNA (Fig 3B). STH assays were performed in over 50 independent experiments, and reproducibility was confirmed.

## A pilot prospective study to evaluate the feasibility of the new SMA screening assay using saliva specimens from SMA patients and carriers

These results provide a framework for innovative SMA-screening assays. In this workflow, the DNA isolation step was omitted, and the evaluation step after PCR was completed in a very short time, requiring only approximately one hour in total (Fig 4A).

We further conducted a prospective clinical study to validate the accuracy of this assay. Five patients with SMA attending the Department of Neurology, Nagoya University Hospital, and two of their parents who were SMA carriers were enrolled in this study. The collected saliva specimens were anonymously randomized and transported to the laboratory of Craif Inc.,

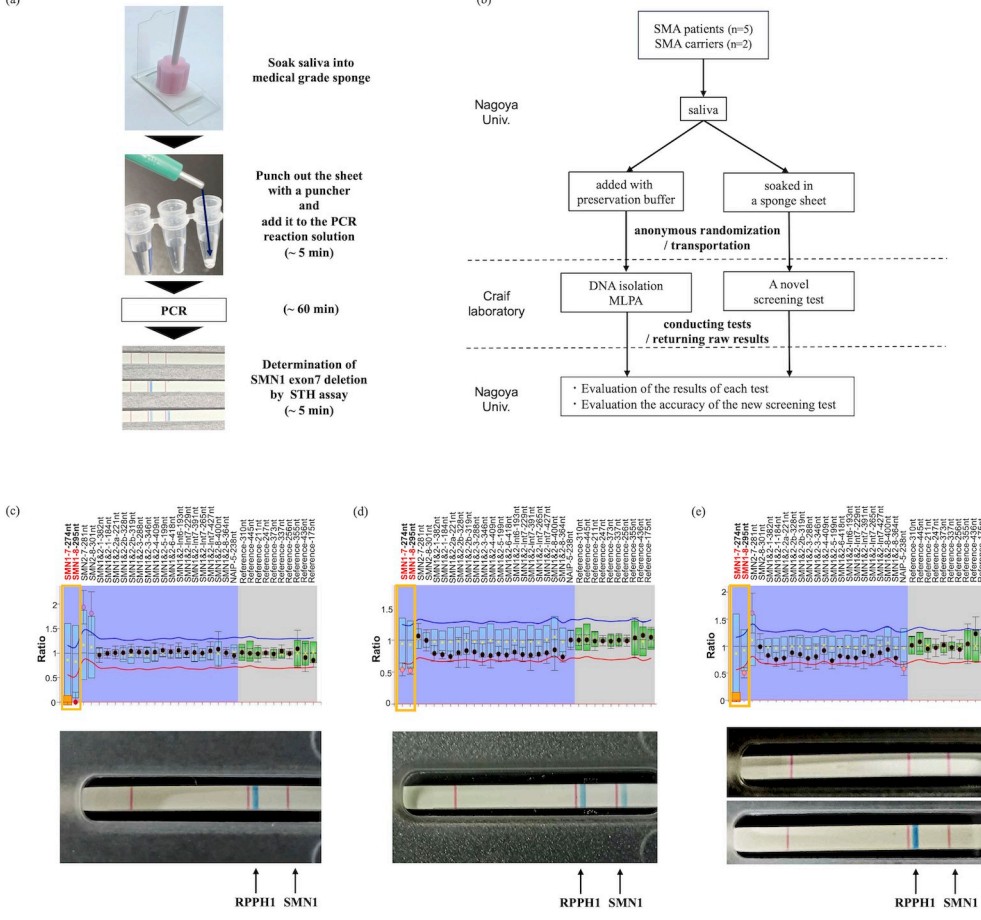

**Fig 4. A pilot prospective study to evaluate the feasibility of the new SMA screening assay.** (a) This rapid and easy-to-use SMA screening assay was made possible by developing new *SMN1*-specific primers with high amplification efficiency of *SMN1*. (b) Clinical study flowchart for evaluating the accuracy of the new SMA screening assay. (c) and (d) Results of the MLPA assay using extracted gDNA (top) and our new assay using the corresponding raw saliva on a sponge sheet (bottom). (c: SMA patient; d: SMA carrier). The area in the yellow box shows the copy number ratio for *SMN1* exons 7 and 8. (e) Results of the MLPA assay using extracted gDNA (top), our new assay using the corresponding raw saliva in the sponge sheet (middle), and saliva with preservation solution (bottom). In the middle panel, neither the *SMN1* band nor the control *RPPH1* band were confirmed, indicating an insufficient sample volume.

where each specimen was analyzed for *SMN1* exon 7 status by our new assay using raw saliva and by the MLPA assay using gDNA extracted from saliva. Each result was analyzed and linked to patient information by collaborators at the Department of Neurology, Nagoya University Hospital, to evaluate the consistency between the assays (Fig 4B). In all cases, the results of our new assay were consistent with the results of the MLPA assay, although saliva added with preservation buffer (total 1 μL) was used for our assay in one case as the amount of saliva soaked in a sponge sheet was insufficient (Fig 4C–4E).

## Discussion

With the advent of new treatments for SMA in recent years [4–7], the importance of SMA screening has become unquestionably vital. Particularly in patients with severe forms of SMA, even a one-week delay in starting treatment can affect prognosis; therefore, identifying patients with SMA in the early postnatal period and linking them to treatment is a major challenge [8].

Although MLPA, digital PCR, and next-generation sequencing-based tests have recently been reported to have a high sensitivity and specificity [17–19], the high cost of a single test and the high level of expertise required to interpret the results are barriers to their use for screening purposes. Quantitative PCR-based assays are widely used as screening methods for SMA, as they are less expensive and easier to perform than other assays [13, 20]. However, the initial capital investment and the need for expertise in interpreting the results make it difficult to implement in ordinary hospitals and clinics, resulting in a significant loss of time before a suspected case of SMA is detected. To solve this problem and to create a system that can provide the fastest and most optimal treatment for patients with SMA, we developed a rapid and easy-to-use screening assay based on conventional non-quantitative PCR.

As previously mentioned, one of the difficulties in developing a screening assay for SMA is the presence of *SMN2*, which has very high homology with the causative *SMN1* gene. We focused on the different base positions (around exon 7) between *SMN1* and *SMN2*, designed forward and reverse primers to contain one of these at (or near) the 3′ end, and screened promising primer sequences. Normally, it is difficult to achieve complete exclusivity, even when allele-specific primers are designed. However, by adding *SMN1* specificity to both the forward and reverse primers, and by searching for appropriate primer lengths, target differential base positions, and PCR protocols, it was possible for the first time to perform a PCR reaction that is completely specific for *SMN1* while maintaining high amplification efficiency, which could not be achieved with previously reported primers for the quantitative PCR of *SMN1*. The conventional PCR system is extremely inexpensive compared to the aforementioned devices, reagents are easy to prepare, and reactions can be completed in approximately an hour, making it very attractive to both the tester and the testee.

Electrophoresis of the PCR products is impractical in hospitals and clinics. Therefore, alternative methods that are easier to implement must be considered. Fortunately, through repeated testing, we were able to confirm that the STH method (TBA, Japan) provided results consistent with those obtained by electrophoresis. However, this was only the case when using our *SMN1*-specific primers with high amplification efficiency of *SMN1*. In addition, the widely used SMA screening test system, which uses dried blood spots, involves an interval between the birth of a newborn and the appropriate time for blood collection [9]. To overcome this problem and develop a less invasive test, a direct PCR method using raw saliva was investigated and was found to be very successful.

One of the limitations of this study is that the incidence of SMA in Japan is relatively low [2], and the clinical trial was conducted using saliva samples from only five SMA patients and two SMA carriers. Nevertheless, the newly developed assay achieved 100% sensitivity, 100%

specificity, and 100% positive predictive value in this investigation. In addition, the reproducibility of the assay was satisfactory. We are currently conducting a large-scale prospective study in newborns.

Another concern is that the effects of mother-derived tissue contamination from breast milk and other sources on the assay could not be excluded. However, this was not expected to pose a major problem, as we demonstrated that gDNA from SMA patients mixed with various ratios of gDNA from healthy individuals did not affect the results when there was up to 10% contamination of normal gDNA, which was equivalent to 20% contamination of gDNA from SMA carriers (S2 Fig). However, the saliva collection method needs to be improved so that a sufficient amount of saliva specimens can be easily and stably introduced into the reaction system. Furthermore, if this test, which can screen for SMA within approximately one hour of birth, is implemented in society, it will be necessary to discuss and negotiate with various experts, including physicians and genetic counselors.

In conclusion, we developed an innovative, inexpensive, and easy-to-use SMA screening assay based on the development of the *SMN1* extremely specific primers with high amplification efficiency. This will certainly lead to a society where patients with SMA can receive disease-modifying therapies within the shortest possible time, thereby conferring the greatest efficacy.

## Supporting information

**S1 Fig. Schematic overview of the *SMN*-coding genes, the primers and probe used in this study.** The five different nucleotides between the *SMN1* and *SMN2* exon7 peri-regions are shown (*SMN1*>*SMN2*). Both forward primers shown here target c.840 cite, while the reverse primer designed in this study also targets c.888+100 cite. The red-letter bases of the primers are complementary to the different bases between the two genes.
(TIFF)

**S2 Fig. STH assay results using the same sample set as in Fig 2C.** From left to right, the mixing ratios of gDNA(*SMN1* CN 0) and gDNA(*SMN1* CN2) are 1:1, 5:1 and 10:1.
(TIFF)

**S1 File. Data for all primers newly designed and evaluated in this study.** Sequence information, target bases, *SMN1* specificity, and Ct values against the control gene are shown for all primers.
(XLSX)

**S1 Raw images. All raw gel images in this study.** All raw gel images in the main and supplemental figures of the manuscript are shown, with the loading order, the identity of the experimental samples, and the method used to capture the images.
(PDF)

## Acknowledgments

The authors greatly appreciate the contributions of all the patients and their parents to this study.

## Author Contributions

**Conceptualization:** Masaki Hirano, Yuki Ichikawa, Atsushi Natsume.

**Data curation:** Masaki Hirano, Kentaro Sahashi.

**Formal analysis:** Masaki Hirano, Kentaro Sahashi.

**Methodology:** Masaki Hirano, Atsushi Natsume.

**Project administration:** Masahisa Katsuno, Atsushi Natsume.

**Resources:** Kentaro Sahashi, Yuki Ichikawa, Masahisa Katsuno.

**Supervision:** Masahisa Katsuno, Atsushi Natsume.

**Visualization:** Masaki Hirano.

**Writing – original draft:** Masaki Hirano.

**Writing – review & editing:** Kentaro Sahashi, Yuki Ichikawa, Masahisa Katsuno, Atsushi Natsume.

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
