## [Decision Letter · Decision Letter 0]

4 Jun 2024

PONE-D-24-10271A rapid and easy-to-use spinal muscular atrophy screening tool based on primers with high specificity and amplification efficiency for SMN1 combined with single-stranded tag hybridization assayPLOS ONE

Dear Dr. Hirano,

Thank you for submitting your manuscript to PLOS ONE. After careful consideration, we feel that it has merit but does not fully meet PLOS ONE’s publication criteria as it currently stands. Therefore, we invite you to submit a revised version of the manuscript that addresses the points raised during the review process.

The manuscript was reviewed by a reviewer and myself. Overall, the experiments are well executed and the data analysis is sound.

Nevertheless, a few points should be addressed:

The control primers utilized (ref 13) should be better defined (is just the same primer sequence or also the fluorophores). In addition, there should be a graphic representation of the genes analyzed and the approximate location of the primers selected and control primers.

The authors should better specify what conditions have been used to run the agarose gels. 100V alone do not define the size of the gel, V x cm is usually   the unit utilized to express the voltage utilized to run an agarose gel.

Figure 4 is difficult to read and should be enlarged or modified, especially the PCR in (a) and panels c,d,e

STH assays in Figures 3 and 4 should have an enhanced contrast/luminosity to clearly visualize the bands.

We look forward to receiving your revised manuscript.

Kind regards,

Massimo Caputi, PhD

Academic Editor

PLOS ONE

“This study was funded by Craif Inc.”

“I have read the journal's policy and the authors of this manuscript have the following competing interests:

Y.I. is a Board member of Craif and owns stock of Craif.

M.H. is an employee of Craif and owns stock options of Craif.

A.N. has a conflict of interest because he received an honorarium for academic guidance from Craif.

Craif is currently applying for a patent for the deliverables developed in this study.”

Please respond by return email with your amended Competing Interests Statement and we will change the online submission form on your behalf.

5. Please upload a copy of Supporting Information Figure (S1 Fig.) which you refer to in your text on page 16.

Reviewers' comments:

Reviewer's Responses to Questions

**Comments to the Author**

1. Is the manuscript technically sound, and do the data support the conclusions?

Reviewer #1: Partly

2. Has the statistical analysis been performed appropriately and rigorously? 

Reviewer #1: Yes

3. Have the authors made all data underlying the findings in their manuscript fully available?

Reviewer #1: Yes

4. Is the manuscript presented in an intelligible fashion and written in standard English?

Reviewer #1: Yes

5. Review Comments to the Author

Reviewer #1: Early diagnosis of Spinal Muscular Atrophy is crucial, as early treatment can drastically improve prognosis. Currently, screens in newborns have a several day turnaround time. The authors claim to have developed a novel, rapid, and non-invasive genetic screening assay for Spinal Muscular Atrophy, designed to be used on newborns shortly after birth, returning results in about an hour.

I think there are some a couple of major revisions that need to get made prior to the manuscript being accepted for publication.

Major:

Through out the manuscript the primers/probe set designed by the authors were compared to screening primers used in reference 13, from this point forward referenced as German primers/probe set. The authors made no mention of any changes they may have made to the German primers, suggesting that no changes were made or they were just no mentioned in the manuscript. This presents a problem because the German probe contains Rhodamine 6 fluorophore which has extensive crosstalk with the NED fluorophore on their RPPH1 probe which could skew results and make them difficult to quantify.

Additionally, in figure 2b, the authors comparing simultaneous amplification of their SMN1 primer set and RPPH1 with the German SMN1 primer set and RPPH1. There is only a 20bp difference between German SMN1 PCR product and RPPH1, is a 2% agarose gel with a 25 minute run time enough to resolve these two bands in the same well.

Minor:

Figure 3a is small, dimly lit and unclear.

Figure 4 c-e (upper) are illegible

I think it's also necessary to include any changes you made to the German primer/probe set used in reference 13. This includes changes made to the probe's fluorophore and the specific tags used for the STH assay

6. PLOS authors have the option to publish the peer review history of their article (what does this mean?). If published, this will include your full peer review and any attached files.

Reviewer #1: No

---

## [Author Response · Author response to Decision Letter 0]

20 Jun 2024

Responses to Reviewers

・We appreciate the reviewers’ comments and suggestions, which have been immensely beneficial to our manuscript. We have corrected our paper in line with the reviewers’ suggestions, which can be observed in our responses below. 

・Regarding references, ref 3 has been updated to the appropriate bibliography as it contained different information than the intended bibliography. In addition, a paper reporting on the STH assay has been added as new ref 14, so the following reference numbers are off by one.

To Academic editor:

1. The control primers utilized (ref 13) should be better defined (is just the same primer sequence or also the fluorophores). In addition, there should be a graphic representation of the genes analyzed and the approximate location of the primers selected and control primers.

>Reply:

We apologize for the inadequate description in the manuscript.

In the Methods section, we have added that the primers for SMN1 with exactly the same sequences as in ref 13 were also used in this study, and that the fluorescent probe (TaqMan MGB probe) with the same sequence as in ref 13 and a different dye (FAM) was used in this study.

We have also added a schematic overview of the genes analyzed and the approximate location of the primers selected and control primers in the new S1 Figure.

2. The authors should better specify what conditions have been used to run the agarose gels. 100V alone do not define the size of the gel, V x cm is usually the unit utilized to express the voltage utilized to run an agarose gel.

>Reply:

We apologize for the confusion caused by using the wrong unit. We have corrected it to V/cm as you indicated.

3. Figure 4 is difficult to read and should be enlarged or modified, especially the PCR in (a) and panels c,d,e. 

>Reply:

We have updated Figure 4 to make it easier to see.

4. STH assays in Figures 3 and 4 should have an enhanced contrast/luminosity to clearly visualize the bands.

>Reply: 

We appreciate you pointing this out to us.

We have re-shot the samples or adjusted the brightness and contrast appropriately to make the blue lines of the STH assay more visible.

To Reviewer #1

1. Throughout the manuscript the primers/probe set designed by the authors were compared to screening primers used in reference 13, from this point forward referenced as German primers/probe set. The authors made no mention of any changes they may have made to the German primers, suggesting that no changes were made or they were just no mentioned in the manuscript. This presents a problem because the German probe contains Rhodamine 6 fluorophore which has extensive crosstalk with the NED fluorophore on their RPPH1 probe which could skew results and make them difficult to quantify.

>Reply:

We apologize for the inadequate description in the manuscript.

In the Methods section, we have added that the primers for SMN1 with exactly the same sequences as in ref 13 were also used in this study, and that the fluorescent probe (TaqMan MGB probe) with the same sequence as in ref 13 and a different dye (FAM) was used in this study.

2. Additionally, in figure 2b, the authors comparing simultaneous amplification of their SMN1 primer set and RPPH1 with the German SMN1 primer set and RPPH1. There is only a 20bp difference between German SMN1 PCR product and RPPH1, is a 2% agarose gel with a 25 minute run time enough to resolve these two bands in the same well.

>Reply:

We appreciate you bringing this very important issue to our attention. To resolve this issue with Figure 2, we performed the same experiments using a new 4% agarose gel and 20 bp ladder (7 V/cm, 60min) and confirmed that the results were the same as in the previous experiments.

3. Figure 3a is small, dimly lit and unclear. Figure 4 c-e (upper) are illegible.

>Reply:

We have updated Figures 3 and 4 to make them more visible.

4. I think it's also necessary to include any changes you made to the German primer/probe set used in reference 13. This includes changes made to the probe's fluorophore and the specific tags used for the STH assay.

>Reply:

In the Methods section, we have added that the primers for SMN1 with exactly the same sequences as in ref 13 were also used in this study, and that the fluorescent probe to detect SMN1/2 (TaqMan MGB probe) had the same sequence as in ref 13 and a different dye (FAM).

We have also added a description of the modifications made to the primers for the STH assay in the Methods section.

---

## [Decision Letter · Decision Letter 1]

18 Jul 2024

A rapid and easy-to-use spinal muscular atrophy screening tool based on primers with high specificity and amplification efficiency for SMN1 combined with single-stranded tag hybridization assay

PONE-D-24-10271R1

Dear Dr. Hirano,

We’re pleased to inform you that your manuscript has been judged scientifically suitable for publication and will be formally accepted for publication once it meets all outstanding technical requirements.

Kind regards,

Massimo Caputi, PhD

Academic Editor

PLOS ONE

Additional Editor Comments (optional):

Reviewers' comments:

Reviewer's Responses to Questions

**Comments to the Author**

1. If the authors have adequately addressed your comments raised in a previous round of review and you feel that this manuscript is now acceptable for publication, you may indicate that here to bypass the “Comments to the Author” section, enter your conflict of interest statement in the “Confidential to Editor” section, and submit your "Accept" recommendation.

Reviewer #1: All comments have been addressed

2. Is the manuscript technically sound, and do the data support the conclusions?

Reviewer #1: Yes

3. Has the statistical analysis been performed appropriately and rigorously? 

Reviewer #1: Yes

4. Have the authors made all data underlying the findings in their manuscript fully available?

Reviewer #1: Yes

5. Is the manuscript presented in an intelligible fashion and written in standard English?

Reviewer #1: Yes

6. Review Comments to the Author

Reviewer #1: The image files for Figures 1-4 look great on their own, however they appear very blurry when inserted into the manuscript. I'm not sure if this is something that will be automatically fixed when the final manuscript is processed, or if action needs to be taken to improve the image quality in the document.

7. PLOS authors have the option to publish the peer review history of their article (what does this mean?). If published, this will include your full peer review and any attached files.

Reviewer #1: No

---

## [Editor Report · Acceptance letter]

22 Jul 2024

PONE-D-24-10271R1 

PLOS ONE

Dear Dr. Hirano, 

I'm pleased to inform you that your manuscript has been deemed suitable for publication in PLOS ONE. Congratulations! Your manuscript is now being handed over to our production team.

Kind regards, 

on behalf of

Dr. Massimo Caputi 

Academic Editor

PLOS ONE